# An Observation of the Microstructure of Cervical Mucus in Cows during the Proestrus, Estrus, and Metestrus Stages and the Impact on Sperm Penetration Ability

**DOI:** 10.3390/vetsci11090391

**Published:** 2024-08-25

**Authors:** Fei Huang, Lu-Lu Zhang, Peng Niu, Xiao-Peng Li, Xue-Yan Wang, Jie Wang, Jie-Ru Wang, Jia-Jia Suo, Di Fang, Qing-Hua Gao

**Affiliations:** 1College of Life Science and Technology, Tarim University, Alar 843300, China; 107572022315@stumail.taru.edu.cn (F.H.); 107572023420@stumail.taru.edu.cn (P.N.); 120240063@taru.edu.cn (J.W.); 107572021314@stumail.taru.edu.cn (J.-R.W.); 107572021317@stumail.taru.edu.cn (J.-J.S.); 2College of Animal Science and Technology, Tarim University, Alar 843300, China; 10757223061@stumail.taru.edu.cn (L.-L.Z.); 2021218318@stumail.taru.edu.cn (X.-P.L.); 2021220424@stumail.taru.edu.cn (X.-Y.W.); 120230093@taru.edu.cn (D.F.); 3Key Laboratory of Tarim Animal Husbandry Science and Technology, Xinjiang Production &Construction Corps, Alar 843300, China

**Keywords:** dairy cows, cervical mucus, estrous period, scanning electron microscopy, sperm penetration tests

## Abstract

**Simple Summary:**

Cervical mucus plays a crucial role in the fertilization process, not only providing energy for sperm but also forming a barrier to block sperm. This experiment investigated the microstructure of cervical mucus and its impact on sperm permeability during the proestrus, estrus, and metestrus in dairy cows. The cervical mucus of cows is mainly composed of four types, and as the estrous cycle changes, the proportion of different types of cervical mucus also changes. There are differences in the microstructure among the four types of cervical mucus. The ability of cervical mucus to allow sperm penetration varies during different estrous phases. These findings provide data supporting the study of cervical mucus.

**Abstract:**

Cervical mucus not only provides energy for sperm but also forms a barrier to block sperm. This paper aims to study the microstructure of cervical mucus in dairy cows during the proestrus, estrus, and metestrus and its effect on sperm permeability. The experiment collected cervical mucus from 60 Holstein cows during these phases, then observed the different shapes of the mucus after crystallization, classified the mucus, and analyzed its proportions. Scanning electron microscopy was used to observe the ultrastructure of the cervical mucus and measure the micro-pore sizes, followed by sperm permeability tests using mucus from different estrous stages and counting the number of permeated sperm. The results indicate that cervical mucus from cows in different estrous phases includes four types (L, S, P, G), with each type constituting a different proportion. During the proestrus, the L type was significantly more prevalent than the other types (*p* < 0.05); during estrus, the S type was significantly more prevalent than the other types (*p* < 0.05); and during the metestrus, the *p* type was significantly more prevalent than the other types (*p* < 0.05). The micro-pore sizes of the same type of cervical mucus did not show significant differences across different estrous phases (*p* > 0.05). However, within the same estrous phase, there were significant differences in the micro-pore sizes among the four types (*p* < 0.05). The number of sperm that permeated the cervical mucus during estrus and metestrus was significantly higher than during the proestrus (*p* < 0.05). This study provides data support for the research on cervical mucus in dairy cows.

## 1. Introduction

Cervical mucus is a constantly changing fluid mainly composed of water, lipids, cholesterol, carbohydrates, inorganic ions, proteins, and other components, with a microstructure resembling a “mesh” [1]. The size of the “mesh” is regulated by the changes in estrogen and progesterone around the time of ovulation, to facilitate sperm passage through the reproductive tract during the optimal period [2,3]. The hydrodynamic properties of cervical mucus impede sperm passage through the cervix during the luteal phase, when it is present in minimal quantities, characterized by high viscosity and opacity [4,5]. However, at ovulation, the production of cervical mucus increases, transforming it into a highly hydrated liquid, with the maximum permeability for sperm at this stage [6,7]. In human research, observations of cervical mucus can be used to diagnose certain reproductive tract diseases and to determine the precise phase of an individual’s physiological cycle [8,9,10]. In dairy farming, cows also exude cervical mucus during estrus, but there is relatively little research on cervical mucus in cows. The microstructure of cervical mucus and its efficiency in allowing sperm passage during different phases of estrus in cows are worthy of study. The estrous cycle of dairy cows is primarily divided into four phases: the proestrus phase, the estrus phase, the metestrus phase, and the diestrus phase [11]. Cows exhibit mounting behavior during estrus, with ovulation typically occurring within 10–15 h after the initiation of mounting behavior [12]. Accurately determining the timing of ovulation in dairy cows and conducting mating or artificial insemination at the exact time can effectively improve the fertilization rate of dairy cows.

The crystalline pattern (dendritic) of cervical mucus can be utilized to detect estrus [13,14]. Cervical mucus is primarily classified into four types: Type L, Type S, Type P, and Type G, these four types of cervical mucus do not exist in isolation but are mixed in different combinations during different periods [15,16]. The different levels of estrogen and progesterone at various stages result in changes in the amount of each type, thereby causing the cervical mucus to differ in its final composition [2]. Type L cervical mucus begins to be secreted 5 days before ovulation, with a microstructure pore size ranging from 0.4 to 3 μm [15,16]. Since the diameter of the sperm head is between 4 and 7 μm [17], cervical mucus of the L-type can, to some extent, inhibit the entry of sperm [15,16]. The crystalline state of Type L mucus presents a shape with a straight or curved central axis, with branches extending at a 90° angle; Type S cervical mucus is secreted around the time of ovulation, with a microstructure pore size ranging from 1.5 to 7 μm, forming a vast “highway” through which sperm can swim smoothly once they come into contact with Type S cervical mucus. The crystalline state of Type S mucus features parallel arranged crystals with short and prominent branches; Type P cervical mucus is secreted at the end of the cervix, with a microstructure pore size between that of Type L and Type S, ranging from 0.4 to 4 μm, serving a filtering and energy-providing function for sperm [15,16]. The crystalline state of Type P mucus presents two basic axes with a 90° angle between them; Type G cervical mucus often appears at various stages of infertility and can form a mucus plug in the cervix, thereby sealing the cervix and preventing sperm from entering the uterus [15,16]. Type G cervical mucus also functions to protect the reproductive tract from external contamination [15,16]. The majority of Type G cervical mucus is coral-like in appearance, without a fixed shape [15,16]. Whether the same pattern of cervical mucus changes exists in dairy cows is worthy of study.

Cervical mucus has different microstructures at different times, which can act as a special barrier for sperm, efficiently filtering sperm [18]. This experiment selected cervical mucus from dairy cows at different time points during estrus to observe the microstructure of the cervical mucus and conduct sperm penetration experiments. The results indicate that the microstructure of cervical mucus from dairy cows during different phases of estrus shows regular changes, and the number of penetrating sperm varies with the changes in the estrous cycle. This study investigates the microstructure of cervical mucus and sperm penetration ability at different time points during the estrous period in dairy cows, providing support for the identification of ovulation, and the timing of artificial insemination in dairy cows.

## 2. Materials and Methods

### 2.1. Ethics Statement

All animal experiments were conducted in accordance with the “Regulations and Guidelines for the Management of Experimental Animals” established by the Ministry of Science and Technology (Beijing, China, 2020 revision). This study was approved by the Institutional Animal Care and Use Committee of Tarim University, Xinjiang, China (protocol code DWBH20220101; approval date: 1 January 2022).

### 2.2. Materials

Cervical Mucus: A total of 60 Holstein cows aged 3–5 years with similar body conditions and reared in the same environment were selected. The cows selected are multiparous dairy cows that have previously undergone artificial insemination. During the trial period, these cows were all in a non-pregnant state. Cervical mucus samples were collected on the day before estrus (proestrus), the day of estrus (estrus), and the day after estrus (metestrus) from each cow (all cows exhibited natural estrus). The collection of cervical mucus was performed at 24 h intervals, as illustrated in Figure 1 (samples were collected on day 21 of the previous estrous cycle, and on day 1 and day 2 of the current estrous cycle).

Semen: Commercially available conventional frozen-thawed semen was used. Commercially available frozen semen is stored in liquid nitrogen. When thawing semen, remove the frozen semen from liquid nitrogen and quickly place it into a 38 °C water bath. Gently shake it for 30 s, then test the vitality and density, and utilize semen with a vitality rate exceeding 0.8 (the sperm concentration of the semen in the cryostraw is 150 million/mL).

Main Reagents and Instruments: A total of 2.5% glutaraldehyde, acetone, optical microscope, critical point drying apparatus, scanning electron microscope, sperm counting chamber, glass slides, and semen straws (0.25 mL per straw).

### 2.3. Collection of Cervical Mucus

The collection of cervical mucus from dairy cows during different phases of estrus was performed by gently pressing the cervix and vagina through the rectum. The cows were first restrained, and the feces within the rectum were cleared. Subsequently, the anus and vulva were cleaned. The collector gently pressed the cervix and vagina through the rectum, repeatedly applying pressure from the cervix towards the vaginal opening until the cervical mucus flowed out. The mucus was then collected using a clean beaker. If cows with uterine inflammation were found during sample collection, they were immediately discarded, and only samples with clear and transparent cervical mucus were collected.

### 2.4. Observation of Cervical Mucus Crystallization

The observation of cervical mucus crystallization in dairy cows during different estrous phases involved initially pipetting 20 μL of cervical mucus onto a glass slide and spreading it evenly. The slide was then air-dried at room temperature for 30 min to ensure that the cervical mucus on the slide was in a dried crystalline state. An optical microscope was used to observe the air-dried cervical mucus. The crystallization patterns of cervical mucus during different estrous phases in cows were analyzed using the model proposed by Mikaela Menarguez [16]. Statistical analysis was performed on the quantity of each type of cervical mucus. For each sample, the five fields of view were randomly selected for observation, and each sample was repeated at least three times. This study involved the simultaneous observation by 5 individuals and their evaluation of the samples.

### 2.5. Scanning Electron Microscopy Was Used to Observe the Microstructure of Cervical Mucus in Cows at Different Stages of Estrus

Cervical mucus samples from different estrous phases were immediately fixed in 2.5% glutaraldehyde solution at 4 °C for 48 h after collection. Following fixation, the samples were washed twice in 0.25 M cacodylate buffer for 30 min each to remove the glutaraldehyde. Subsequently, the samples were dehydrated through a series of concentrations of acetone (30, 50, 70, 90, 100% *v*/*v*), with each concentration of acetone used for 2 h of dehydration. The samples were then dried using a Leica EM CPD300 automatic critical point dryer (Leica EM CPD300, Leica, Germany), metallized with a Cressington 108Auto high-performance automated sputter coater (Cressington 108Auto, Cressington, UK), and finally observed with a Thermo Scientific Apreo S scanning electron microscope (Apreo, Thermo Fisher Scientific, USA). Statistical analysis was conducted on the micro-pore sizes of the cervical mucus, with each sample repeated at least three times. This study involved the simultaneous observation by 5 individuals and their evaluation of the samples.

### 2.6. Study on the Sperm Penetration Ability of Cervical Mucus in Dairy Cows during Different Estrous Phases

The sperm penetration experiments were refined based on the work of Tang, S et al [19]. Sperm penetration experiments were conducted using empty semen straws (as shown in Figure 2); with an available length of 10 cm for each straw, the capacity is 200 μL. To facilitate measurement, a mark was made at 1 cm from the opening of the semen straw using a marker pen. Cervical mucus was first aspirated and filled into the straw up to the 1 cm mark from the opening (180 μL cervical mucus). Then, semen was added with a motility rate exceeding 0.8 into each straw, adding approximately 20 μL of semen per frozen semen straw at a concentration of 150 million/mL. The filled semen straw was placed horizontally in a 37 °C, 5% CO_2_ incubator for 30 min. Subsequently, the straw was cut directly at the midpoint, and the mucus from the end, along with the penetrating sperm, was transferred into a 1.5 mL centrifuge tube. This was then centrifuged at a speed of 300 g/min for a duration of 5 min; the supernatant was subsequently discarded, and the remaining 20 μL of liquid was gently homogenized through pipetting. Finally, sperm counting was conducted under the microscope using a sperm counting plate, with each sample repeated at least three times. This study involved simultaneous observation by 5 individuals and their evaluation of the samples.

### 2.7. Statistical Analysis

The collection volume and type ratio of cervical mucus were statistically analyzed using SPSS (IBM SPSS 25.0, SPSS Inc., Chicago, IL, USA) statistical analysis software with one-way analysis of variance (ANOVA). The scanning electron microscopy results of the cervical mucus were analyzed using Image J software (https://imagej.net/ij/, accessed on 20 July 2024). For the sperm counts after penetration, the ANOVA method in the SPSS statistical analysis software was used to assess the significance of differences between groups.

## 3. Results

### 3.1. Observation of Cervical Mucus Crystallization in Dairy Cows during Different Estrous Phases Using Optical Microscopy

The quantities collected during each phase are presented in Table 1. During the proestrus phase, cervical mucus was collected from 58 cows; during estrus, from 57 cows; and during the metestrus phase, from 57 cows. The crystallization of the cervical mucus during the proestrus, estrus, and metestrus phases in dairy cows was observed under an optical microscope. The cervical mucus from all three phases included Types L, S, P, and G (as shown in Figure 3), with varying concentrations of each type during proestrus, estrus, and metestrus (as shown in Table 2). In the proestrus phase, Type L mucus (57.36 ± 5.17%) was significantly more prevalent than Type S (19.42 ± 4.33%), Type P (16.92 ± 4.27%), and Type G (3.55 ± 1.09%) (*p* < 0.05). During estrus, Type S mucus (55.82 ± 6.13%) was significantly more abundant than Type L (23.53 ± 4.01%), Type P (18.94 ± 3.25%), and Type G (5.02 ± 1.47%) (*p* < 0.05). In the metestrus phase, Type P mucus (62.35 ± 5.25%) was significantly more prevalent than Type L (13.66 ± 3.62%), Type S (25.41 ± 4.89%), and Type G (4.85 ± 1.57%) (*p* < 0.05).

### 3.2. Scanning Electron Microscopy Observation of Cervical Mucus

The scanning electron microscopy observation of different types of cervical mucus (Types L, S, P, and G) from dairy cows during the proestrus (as shown in Figure 4), estrus, and metestrus phases revealed that the microstructures of the cervical mucus types were consistent across the different estrous phases. The pore sizes of the same type of cervical mucus across different estrous phases showed no significant differences (*p* > 0.05). However, the pore sizes of different types of cervical mucus within the same estrous phase were significantly different (*p* < 0.05), (as shown in Table 3).

### 3.3. Statistical Results of Sperm Counts Penetrating Cervical Mucus in Dairy Cows during Proestrus, Estrus, and Metestrus

The sperm counts that penetrated the cervical mucus collected through sperm penetration experiments were statistically analyzed (as shown in Table 4). The cervical mucus during estrus allowed for more sperm penetration than the cervical mucus during metestrus, although the difference was not significant, with 68 ± 7.32 million and 58 ± 6.61 million penetrating sperms, respectively (*p* > 0.05). However, both the cervical mucus during estrus and metestrus showed significantly more penetrating sperms than the cervical mucus during proestrus, with 36 ± 5.18 million penetrating sperms (*p* < 0.05).

## 4. Discussion

This experiment used an optical microscope to observe the crystallization of cervical mucus with different shapes, obtaining the same cervical mucus crystals as Vojtěch Pešan [13]. It has been discovered for the first time that the cervical mucus of dairy cows also includes four types: L-type, S-type, P-type, and G-type. This experiment used cervical mucus from the proestrus, estrus, and metestrus phases. Cervical mucus types L, S, P, and G were observed in the cervical mucus of all three estrous phases, indicating that the composition of cervical mucus is roughly the same during different estrous periods. Cervical mucus is mainly composed of these four types [20], with the typical fern-like Type L appearing predominantly near the estrus phase [21]. This is crucial for determining the onset of estrus, and it suggests that the crystalline type of cervical mucus may be related to the timing of ovulation [22]. We also analyzed the proportion of each type of cervical mucus during different estrous phases. Type L cervical mucus was relatively more abundant in the proestrus phase; it was found that L-type cervical mucus is relatively more abundant in the proestrus phase; S-type cervical mucus is relatively more abundant during estrus; and P-type cervical mucus is relatively more abundant in the metestrus phase. The occurrence of this result is due to the control of the branch-like distribution of cervical mucus by estrogen and progesterone, two ovarian hormones. The crystallization of cervical mucus occurs under the influence of estrogen, while progesterone reduces the formation of the branch-like pattern. Therefore, it is very useful in predicting the onset of estrus, different stages of estrus, and the timing of ovulation in cattle [23].

This study provides a more comprehensive investigation into bovine cervical mucus. We utilized scanning electron microscopy to observe the microstructure of cervical mucus and conducted pore size analysis, revealing that there is no significant difference in the micro-pore size of the same type of cervical mucus at different stages of estrus. This finding is consistent with the research results of Pilar Vigil and colleagues [15]. We have discovered that there is a significant difference in the micro-pore size of different types of cervical mucus during the same estrous period. Among the four types of cervical mucus, only the S-type cervical mucus has a micro-pore size larger than the width of a sperm head 4~7 μm [15,17], with an average pore size ranging from 4.13~8.75 μm. The pore size of Type P cervical mucus 2.43~4.68 μm was also slightly larger than the width of the sperm head. The microscopic pore size of cervical mucus directly affects the efficiency of sperm penetration. In the sperm penetration experiment, it was found that the highest number of sperm penetrate the cervical mucus during estrus, which is closely related to the pore size of different types of cervical mucus [24]. Type S cervical mucus is more prevalent in the cervical mucus during estrus, and the average microscopic pore size of Type S cervical mucus is slightly larger than the width of the sperm head, facilitating sperm penetration in the cervical mucus [25,26]. The speed at which sperm swim in cervical mucus is approximately 70~110 μm/s [27]. In the design of this experiment, sperm were allowed to swim in the semen capillary for 30 min, providing them with ample time to swim and facilitating the accurate collection of sperm. During the proestrus, estrus, and metestrus phases, the cervical mucus during estrus allowed for the most sperm penetration, which may also be related to the cow’s ovulation during estrus. The reproductive mechanism of cows leads to an increase in Type S cervical mucus during estrus, and the pore size of Type S cervical mucus is the largest among the four types, thereby increasing the likelihood of sperm penetration. The characteristics of cervical mucus undergo changes during the ovulatory cycle. Near ovulation, due to the gelatinous hydration caused by estrogen, the mucus volume increases, becoming less viscoelastic, thus promoting sperm penetration. In contrast, during the luteal phase, under the influence of progesterone, the mucus becomes a low-water content, highly viscoelastic structure, serving as a barrier to sperm [28].

## 5. Conclusions

In this experiment, L-type, S-type, P-type, and G-type cervical mucus were observed in cervical mucus during the proestrus, estrus, and metestrus phases, with different proportions of these four types during different estrous phases. The microscopic pore sizes of L-type, S-type, P-type, and G-type cervical mucus also varied, with S-type cervical mucus having the largest pore size, with an average pore size ranging from 4.13 to 8.75 μm. The sperm penetration effects of cervical mucus during different estrous phases also differed, being closely related to the content of S-type cervical mucus.

## Figures and Tables

**Figure 1 vetsci-11-00391-f001:**
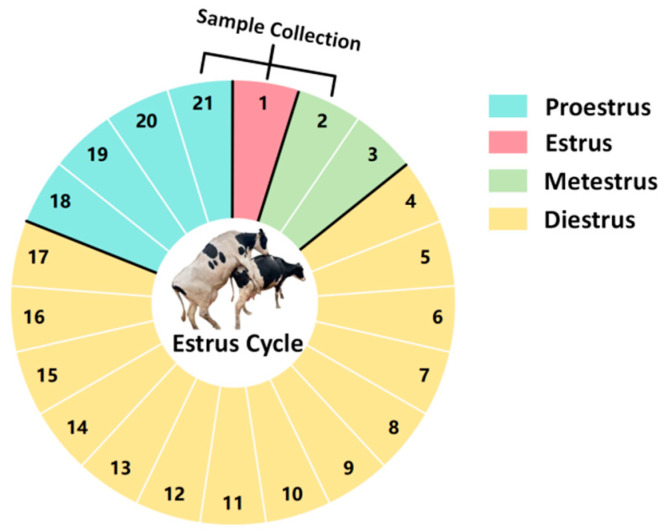
Diagram of the estrous cycle in dairy cows.

**Figure 2 vetsci-11-00391-f002:**
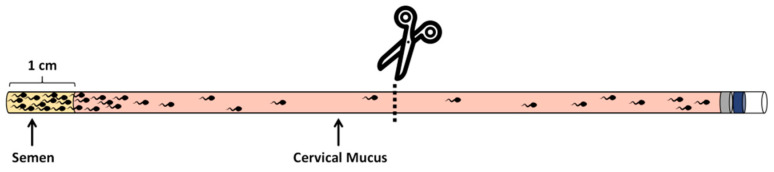
Schematic diagram of sperm penetration experiment.

**Figure 3 vetsci-11-00391-f003:**
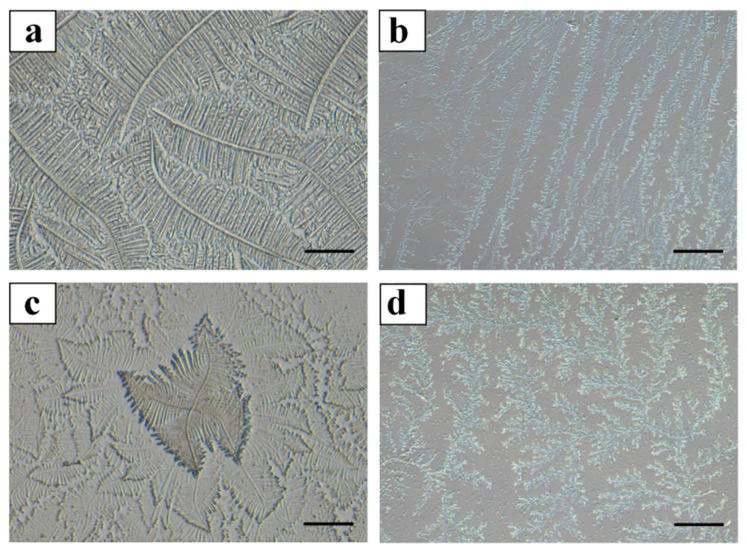
Crystallization patterns of different types of cervical mucus in dairy cows. Note: In Figure 3, panel (**a**) represents Type L cervical mucus, characterized by a straight or curved central axis with branches extending at a 90° angle; panel (**b**) shows Type S cervical mucus, with parallel-aligned crystals featuring short and prominent branches; panel (**c**) depicts Type P cervical mucus, which has two primary axes at a 90° angle to each other; panel (**d**) illustrates Type G cervical mucus, predominantly exhibiting a coral-like structure with no fixed form. Scale bar = 200 μm.

**Figure 4 vetsci-11-00391-f004:**
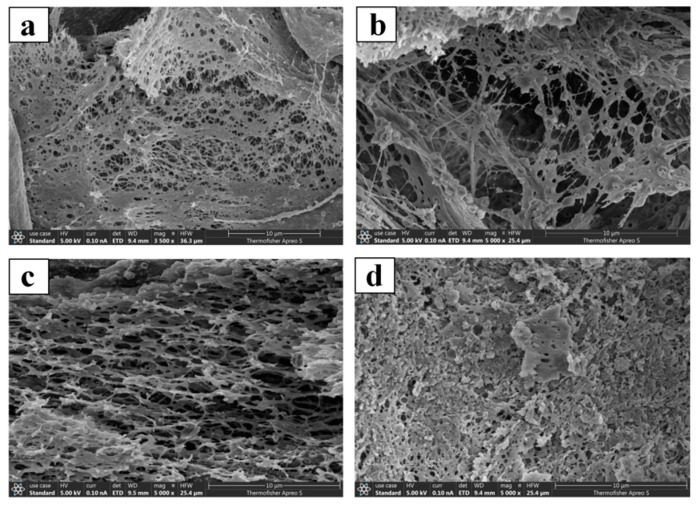
Scanning electron microscopy images of cervical mucus. In Figure 4, panel (**a**) represents Type L cervical mucus; panel (**b**) represents Type S cervical mucus; panel (**c**) represents Type P cervical mucus; panel (**d**) represents Type G cervical mucus.

**Table 1 vetsci-11-00391-t001:** Cervical mucus collection volumes.

Group	Number of Cows (Head)	Cervical Mucus Collection Volume (mL)
Proestrus	60 (58)	32.55 ± 6.31 ^b^
Estrus	60 (57)	59.83 ± 14.45 ^a^
Metestrus	60 (57)	27.36 ± 9.92 ^b^

The number in parentheses is the number of cows from which samples were collected. Values within the same column with the same superscript letters indicate no significant difference (*p* > 0.05), while values with different superscript letters indicate a significant difference (*p* < 0.05).

**Table 2 vetsci-11-00391-t002:** Proportion of different types of cervical mucus in dairy cows during proestrus, estrus, and metestrus.

Group	Number of Cows (Head)	Content of Different Types of Cervical Mucus (%)			
		L	S	P	G
Proestrus	60 (58)	57.36 ± 5.17 ^a^	19.42 ± 4.33 ^bc^	16.92 ± 4.27 ^bc^	3.55 ± 1.09 ^d^
Estrus	60 (57)	23.53 ± 4.01 ^bc^	55.82 ± 6.13 ^a^	18.94 ± 3.25 ^bc^	5.02 ± 1.47 ^d^
Metestrus	60 (57)	13.66 ± 3.62 ^c^	25.41 ± 4.89 ^b^	62.35 ± 5.25 ^a^	4.85 ± 1.57 ^d^

The number in parentheses is the number of cows from which samples were collected. Within the same column, data with the same superscript letters indicate no significant difference (*p* > 0.05), while data with different superscript letters indicate a significant difference (*p* < 0.05).

**Table 3 vetsci-11-00391-t003:** Measurement results of pore sizes of different types of cervical mucus in dairy cows during proestrus, estrus, and metestrus.

Group	Number of Cows (Head)	Pore Length of Different Types of Cervical Mucus (μm)			
		L	S	P	G
Proestrus	60 (58)	1.82 ± 0.75 ^c^	6.53 ± 1.94 ^a^	3.38 ± 0.95 ^bc^	0.38 ± 0.29 ^d^
Estrus	60 (57)	1.75 ± 0.86 ^b^	6.39 ± 2.02 ^a^	3.57 ± 1.05 ^ab^	0.42 ± 0.33 ^c^
Metestrus	60 (57)	1.79 ± 0.72 ^b^	6.44 ± 2.31 ^a^	3.42 ± 1.26 ^ab^	0.45 ± 0.36 ^c^

The number in parentheses is the number of cows from which samples were collected. Within the same column, data with the same superscript letters indicate no significant difference (*p* > 0.05), while data with different superscript letters indicate a significant difference (*p* < 0.05).

**Table 4 vetsci-11-00391-t004:** Sperm penetration results in cervical mucus of dairy cows during proestrus, estrus, and metestrus.

Group	Number of Cows (Head)	Penetrating Sperm Counts (Million/mL)
Proestrus	60 (58)	36 ± 5.18 ^c^
Estrus	60 (57)	68 ± 7.32 ^a^
Metestrus	60 (57)	58 ± 6.61 ^ab^

The number in parentheses is the number of cows from which samples were collected. Within the same column, data with the same superscript letters indicate no significant difference (*p* > 0.05), while data with different superscript letters indicate a significant difference (*p* < 0.05).

## Data Availability

The original contributions presented in the study are included in the article; further inquiries can be directed to the corresponding authors.

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
