# Peer review of "An Observation of the Microstructure of Cervical Mucus in Cows during the Proestrus, Estrus, and Metestrus Stages and the Impact on Sperm Penetration Ability"

_vetsci, 2024, doi:10.3390/vetsci11090391_

Round 1
Reviewer 1 Report
Comments and Suggestions for Authors
The manuscript entitled "Observation of the Microstructure of Cervical Mucus in Cows During the Proestrus, Estrus, and Metestrus Stages and Impact on Sperm Penetration Abilit" presents a well-structured experiment addressing a topic of significant reproductive interest: the fertility of dairy cows. The study is meticulously designed to contribute valuable insights into this crucial area, which holds considerable importance for both the agricultural industry and reproductive science.
However, I must recommend a major revision of the manuscript, especially for several aspects which I will now describe.
General comments
Although the topic is undoubtedly of great interest, the authors do not emphasize the innovative aspects of their study (the sperm penetration test and electron microscopy) and their potential application in improving reproductive parameters in dairy cows. On the other hand, the conclusions must address the practical applications of the results (or note if there are none). This would offer a more comprehensive perspective on the study's relevance and its potential contributions to the field of reproductive science.
Specific comments
It is unfortunate that the study did not include any cows with reproductive pathological conditions. The comparison of cervical mucus between fertile and infertile females (those with reproductive pathologies) could have provided new insights in this field.
Summary-Abstract
-Lines 13-14 and 21-22. Please, clarify “Cervical mucus plays a crucial role in the fertilization process, not only providing energy for sperm”, Could you provide any bibliographic references to support this sentence?
Materials and Methods
-Lines 154-67. First of all, it is necessary to provide information on whether or not the females were inseminated in that heat.
Overall, the procedure is relatively well understood. However, it should be clarified whether this procedure is novel or if it is based on previous studies. Additionally, the method used to quantify the number of spermatozoa that penetrate the mucus remains unclear.
Furthermore, the process of thawing the straw is not specified, nor is it mentioned whether any semen analysis was conducted prior to the experiment. Given that the quality, vitality, and motility of spermatozoa are typically not uniform across all straws, any variation in these parameters could potentially impact the final results. Therefore, such details are crucial for the interpretation of the study's findings.
Results
As a suggestion, creating a drawing or diagram based on the photographs (either from slides or electron microscopy) could be highly beneficial for future studies, as it may serve as a valuable reference.
Lines 260-5. It is important not only to clarify how these concentration results were obtained, but also to conduct a more in-depth analysis of sperm vitality, motility, DNA integrity, and other relevant parameters. This additional analysis would be particularly valuable, as it is likely that only the "best" spermatozoa were able to penetrate the mucus. Such insights could significantly enhance the understanding of the selection process and the overall findings of the study.
Discussion
As for the discussion, it is quite broad and well argued, but I still do not see the novelties and practical contribution of the work highlighted.
Conclusion
See general comments

Author Response
Comment 1: [Summary-Abstract -Lines 13-14 and 21-22. Please, clarify “Cervical mucus plays a crucial role in the fertilization process, not only providing energy for sperm”, Could you provide any bibliographic references to support this sentence?]
Response 1: We appreciate the reviewer’s concern. We agree with this comment. [As this is situated within the abstract section, I regret my inability to directly integrate citations into the manuscript. However, I shall append pertinent references concerning the assertion "Cervical mucus plays a pivotal role in the fertilization process, not solely serving as an energy source for sperm" below, with the 18th reference in the manuscript being germane. The references are as follows:
- Patricio M,; Mauricio, M.R,; Pilar, V. Human cervical mucus: relationship between biochemical characteristics and ability to allow migration of spermatozoa. Human Reproduction 1993, 8, 78-83. doi:10.1093/oxfordjournals.humrep.a137879
- Schimpf, U.; Caldas-Silveira, E.; Katchan, L.; Vigier-Carriere, C.; Lantier, I.; Nachmann, G.; Gidlof, S.; Jonasson, A.F.; Bjorndahl, L.; Trombotto, S., et al. Topical reinforcement of the cervical mucus barrier to sperm. Sci Transl Med 2022, 14, m2417, doi:10.1126/scitranslmed.abm2417.
- Lorraine, R.; Hanrahan, J.P.; Tharmala, T., et al. Cervical mucus sialic acid content determines the ability of frozen-thawed ram sperm to migrate through the cervix. Reproduction. 2019, 157, 259-271. doi:10.1530/rep-18-0547]
Comment 2: [Materials and Methods -Lines 154-67. First of all, it is necessary to provide information on whether or not the females were inseminated in that heat.]
Response 2: We appreciate the reviewer’s comment. We agree with this comment. [Following your comment, we have added the incubator temperature during sperm penetration. The revised content can be seen in line 163 of the revised manuscript.]
Comment 3: [Overall, the procedure is relatively well understood. However, it should be clarified whether this procedure is novel or if it is based on previous studies. Additionally, the method used to quantify the number of spermatozoa that penetrate the mucus remains unclear.]
Response 3: We appreciate the reviewer’s concern. We agree with this comment. [This procedure is an improvement based on the work of Stephen Tang and others, and we have added the relevant reference in the manuscript. Additionally, we approach to quantifying sperm count is microscopic observation and enumeration using a sperms counting plate, which has been detailed in the manuscript. The reference is as follows:
- Tang, S.; Garrett, C.; Baker, H.W. Comparison of human cervical mucus and artificial sperm penetration media. Hum Reprod 1999, 14, 2812-2817, doi:10.1093/humrep/14.11.2812.
The revised content can be seen in lines 160-161, lines169-170 of the revised manuscript.]
Comment 4: [Furthermore, the process of thawing the straw is not specified, nor is it mentioned whether any semen analysis was conducted prior to the experiment. Given that the quality, vitality, and motility of spermatozoa are typically not uniform across all straws, any variation in these parameters could potentially impact the final results. Therefore, such details are crucial for the interpretation of the study's findings.]
Response 4: We appreciate the reviewer’s concern. We agree with this comment. [We have added the method for thawing straws of frozen semen in the manuscript. Before the experiment, we conducted semen analysis and ensured that each straw of frozen semen contained more than 20 million effective sperm, which has been included in the manuscript. The revised content can be seen in lines 114-116 of the revised manuscript.]
Comment 5: [As a suggestion, creating a drawing or diagram based on the photographs (either from slides or electron microscopy) could be highly beneficial for future studies, as it may serve as a valuable reference.]
Response 5: We appreciate the reviewer’s comment. [Since the results obtained from the experiment seem to follow a certain pattern, we are preparing to write a related patent. I’m sorry, the results not shown in the manuscript cannot be used to create a chart.]
Comment 6: [Lines 260-5. It is important not only to clarify how these concentration results were obtained, but also to conduct a more in-depth analysis of sperm vitality, motility, DNA integrity, and other relevant parameters. This additional analysis would be particularly valuable, as it is likely that only the "best" spermatozoa were able to penetrate the mucus. Such insights could significantly enhance the understanding of the selection process and the overall findings of the study. ]
Response 6: We appreciate the reviewer’s comment. We agree with this comment. [We have added the counting of sperm quantity before penetration, semen quality testing, as well as the collection of semen and sperm counting after penetration to the manuscript. The revised content can be seen in lines 116-118、 lines 166-168、lines 171-174 of the revised manuscript. Currently, we have not conducted tests for sperm DNA integrity, but we plan to test these aspects of sperm DNA integrity in later experiments.]
Comment 7: [As for the discussion, it is quite broad and well argued, but I still do not see the novelties and practical contribution of the work highlighted. ]
Response 7: We appreciate the reviewer’s concern. [We have made slight modifications to the discussion section to highlight its novelties and practical contribution. The revised content can be seen in lines 299-300, lines 309-312, lines 328-331 of the revised manuscript.]
Reviewer 2 Report
Comments and Suggestions for Authors
Congratulations for this huge and very well introduced and described work.I have had many pleasure to read it. These observations offer new perspectives to a better choice of te insemination time. It could be intertesting in the future to analyze the effects of microstructure on fertility.
59 I disagree with with this sentence « Cows typically ovulate during the estrus phase » Oestus phase can be defined by the period of mounted activity. So ovulation appears 10 t 15 hours after the end of this period i.e. during metoestrus.
64 could you define the signification of these four letters L S P G
71 and 82 both type of mucus (L and G) are preventing sperm from entering the uterus : can you explain ?106 how can you exclude that the mucus was free of infection or contamination ? Can we imagine that such alterations can induce changes of mucus rheology ?
131 how many people were involved in the observation of mucus ?
152 It coud be interesting to identify a micropore on the electronic microscope picture
153 could you describe the method of micropores measurements ? How many micropores have you counted by sampling ?
306 you mention a head spermatozoa diameter of 5 to 7 μm but 4 to 6 μm in the introduction (line 70)
307 supress the bracketts
Author Response
Comment 1: [59 I disagree with with this sentence « Cows typically ovulate during the estrus phase » Oestus phase can be defined by the period of mounted activity. So ovulation appears 10 t 15 hours after the end of this period i.e. during metoestrus.]
Response 1: We thank the reviewer for noting our mistake. We agree with this comment. Following your comment, [we have changed the sentence to "Cows exhibit mounting behavior during estrus, with ovulation typically occurring within 10-15 h after the initiation of mounting behavior". The revised content can be seen in lines 59-61 of the revised manuscript.]
Comment 2: [64 could you define the signification of these four letters L S P G]
Response 2: We appreciate the reviewer’s comment. We agree with this comment. [we have changed the sentence to “Cervical mucus is primarily classified into four types: Type L: Type-loaf cervical mucus; Type S: Type-string cervical mucus; Type P: Type-peak cervical mucus; Type G: Type-gestagenic cervical mucus, these four types of cervical mucus do not exist in isolation but are mixed in different combinations during different periods.” The revised content can be seen in lines 65-67 of the revised manuscript.]
Comment 3: [71 and 82 both type of mucus (L and G) are preventing sperm from entering the uterus : can you explain ?]
Response 3: We appreciate the reviewer’s comment. [The G-type cervical mucus appears in various stages of infertility, thus the G-type can impede the entry of sperm, as the microscopic pores of G-type cervical mucus are small, insufficient to allow sperm passage. The micro-porosity of L-type cervical mucus is also relatively small, capable of inhibiting the entry of sperm to a certain extent. I have revised the sentences in the manuscript to "Cervical mucus of the L-type can to some extent inhibit the entry of sperm". The revised content can be seen in lines 72 of the revised manuscript.]
Comment 4: [106 how can you exclude that the mucus was free of infection or contamination ? Can we imagine that such alterations can induce changes of mucus rheology ?]
Response 4: We appreciate the reviewer’s comment. We agree with this comment. [During the sample collection process, we performed cleansing and disinfection of the external genitalia of cows to prevent contamination. If cows with uterine inflammation were found during sample collection, they were immediately discarded, and only samples with clear and transparent cervical mucus were collected. Ensure uniformity in the collection of cervical mucus. We have added the sample collection requirements into the manuscript. The revised content can be seen in lines 134 – 136 of the revised manuscript.]
Comment 5: [131 how many people were involved in the observation of mucus ?]
Response 5: We appreciate the reviewer’s concern. [This study involved the simultaneous observation by 5 individuals and their evaluation of the samples, and have added it into the manuscript. The revised content can be seen in lines 148-149, lines 162-164, lines 182-183 of the revised manuscript.]
Comment 6: [152 It coud be interesting to identify a micropore on the electronic microscope picture.]
Response 6: We appreciate the reviewer’s comment. [Identifying micropores on electron microscope images is indeed an intriguing method.]
Comment 7: [153 could you describe the method of micropores measurements ? How many micropores have you counted by sampling ?]
Response 7: We appreciate the reviewer’s concern. [We analyze the micropores using the Image J software, importing the images obtained from the experiment into the software and utilizing the pore size analysis function to obtain the average pore size of each image. We randomly capture three fields of view for each parameter of every sample for analysis.]
Comment 8: [306 you mention a head spermatozoa diameter of 5 to 7 μm but 4 to 6 μm in the introduction (line 70)]
Response 8: We thank the reviewer for noting our mistake. [This is our writing mistake, we have changed “4-6μm” to “4-7μm” and added the reference. The revised content can be seen in lines 71 – 72 of the revised manuscript.]
Comment 9: [307 supress the bracketts.]
Response 9: We appreciate the reviewer’s comment. [We have removed the parentheses. The revised content can be seen in lines 324 – 325 of the revised manuscript.]
Round 2
Reviewer 1 Report
Comments and Suggestions for Authors
I would like to express my gratitude to the authors for providing clarifications, both in their responses and within the text. However, I continue to believe there is a lack of thorough semen analysis in the samples. I am aware that addressing this issue may no longer be feasible within the confines of the current study. Nonetheless, I encourage the authors to consider incorporating this aspect into their future research.
Furthermore, the authors have not included information on whether the females were inseminated and whether they became pregnant (if insemination was performed). This information is crucial and should be included in the article.
Author Response
Comment 1: [I would like to express my gratitude to the authors for providing clarifications, both in their responses and within the text. However, I continue to believe there is a lack of thorough semen analysis in the samples. I am aware that addressing this issue may no longer be feasible within the confines of the current study. Nonetheless, I encourage the authors to consider incorporating this aspect into their future research.]
Response 1: We appreciate the reviewer’s concern. [We will incorporate a thorough analysis of penetrating semen into our next research plan.]
Comment 2: [Furthermore, the authors have not included information on whether the females were inseminated and whether they became pregnant (if insemination was performed). This information is crucial and should be included in the article.]
Response 2: We appreciate the reviewer’s comment. We agree with this comment. [Following your comment, We have added descriptions of the cows’ pregnancy status and whether they have been artificially inseminated in the manuscript. The revised content can be seen in lines 109-111 of the revised manuscript.]